# Towards Understanding Label Smoothing

## Abstract

Label smoothing regularization (LSR) has a great success in training deep neural networks by stochastic algorithms such as stochastic gradient descent and its variants. However, the theoretical understanding of its power from the view of optimization is still rare. This study opens the door to a deep understanding of LSR by initiating the analysis. In this paper, we analyze the convergence behaviors of stochastic gradient descent with label smoothing regularization for solving non-convex problems and show that an appropriate LSR can help to speed up the convergence by reducing the variance. More interestingly, we proposed a simple yet effective strategy, namely **T**wo-**S**tage **LA**bel smoothing algorithm (TSLA), that uses LSR in the early training epochs and drops it off in the later training epochs. We observe from the improved convergence result of TSLA that it benefits from LSR in the first stage and essentially converges faster in the second stage. To the best of our knowledge, this is the first work for understanding the power of LSR via establishing convergence complexity of stochastic methods with LSR in non-convex optimization. We empirically demonstrate the effectiveness of the proposed method in comparison with baselines on training ResNet models over benchmark data sets.

## 1 Introduction

In training deep neural networks, one common strategy is to minimize cross-entropy loss with one-hot label vectors, which may lead to overfitting during the training progress that would lower the generalization accuracy (Müller et al., 2019). To overcome the overfitting issue, several regularization techniques such as $\ell_1$-norm or $\ell_2$-norm penalty over the model weights, Dropout which randomly sets the outputs of neurons to zero (Hinton et al., 2012b), batch normalization (Ioffe & Szegedy, 2015), and data augmentation (Simard et al., 1998), are employed to prevent the deep learning models from becoming over-confident. However, these regularization techniques conduct on the hidden activations or weights of a neural network. As an output regularizer, label smoothing regularization (LSR) (Szegedy et al., 2016) is proposed to improve the generalization and learning efficiency of a neural network by replacing the one-hot vector labels with the smoothed labels that average the hard targets and the uniform distribution of other labels. Specifically, for a $K$-class classification problem, the one-hot label is smoothed by $\mathbf{y}^{\text{LS}} = (1 - \theta)\mathbf{y} + \theta\widehat{\mathbf{y}}$, where $\mathbf{y}$ is the one-hot label, $\theta \in (0, 1)$ is the smoothing strength and $\widehat{\mathbf{y}} = \frac{1}{K}$ is a uniform distribution for all labels. Extensive experimental results have shown that LSR has significant successes in many deep learning applications including image classification (Zoph et al., 2018; He et al., 2019), speech recognition (Chorowski & Jaitly, 2017; Zeyer et al., 2018), and language translation (Vaswani et al., 2017; Nguyen & Salazar, 2019).

Due to the importance of LSR, researchers try to explore its behavior in training deep neural networks. Müller et al. (2019) have empirically shown that the LSR can help improve model calibration, however, they also have found that LSR could impair knowledge distillation, that is, if one trains a teacher model with LSR, then a student model has worse performance. Yuan et al. (2019a) have proved that LSR provides a virtual teacher model for knowledge distillation. As a widely used trick, Lukasik et al. (2020) have shown that LSR works since it can successfully mitigate label noise. However, to the best of our knowledge, it is unclear, at least from a theoretical viewpoint, how the introduction of label smoothing will help improve the training of deep learning models, and to what stage, it can help. In this paper, we aim to provide an affirmative answer to this question and try to deeply understand why and how the LSR works from the view of optimization.

Our theoretical analysis will show that an appropriate LSR can essentially reduce the variance of stochastic gradient in the assigned class labels and thus it can speed up the convergence. Moreover, we will propose a novel strategy of employing LSR that tells when to use LSR. We summarize the main contributions of this paper as follows.

- It is the **first work** that establishes improved iteration complexities of stochastic gradient descent (SGD) (Robbins & Monro, 1951) with LSR for finding an $\epsilon$-approximate stationary point (Definition 1) in solving a smooth non-convex problem in the presence of an appropriate label smoothing. The results theoretically explain why an appropriate LSR can help speed up the convergence. (Subsection 4.1)

- We propose a simple yet effective strategy, namely **T**wo-**S**tage **LA**bel smoothing (TSLA) algorithm, where in the first stage it trains models for certain epochs using a stochastic method with LSR while in the second stage it runs the same stochastic method without LSR. The proposed TSLA is a generic strategy that can incorporate many stochastic algorithms. With an appropriate label smoothing, we show that TSLA integrated with SGD has an **improved** iteration complexity (Subsection 4.3) and **better** testing error (Subsection 4.4), compared to the SGD with LSR and the SGD without LSR.

## 2 Related Work

In this section, we introduce some related work. A closely related idea to LSR is confidence penalty proposed by Pereyra et al. (2017), an output regularizer that penalizes confident output distributions by adding its negative entropy to the negative log-likelihood during the training process. The authors (Pereyra et al., 2017) presented extensive experimental results in training deep neural networks to demonstrate better generalization comparing to baselines with only focusing on the existing hyper-parameters. They have shown that LSR is equivalent to confidence penalty with a reversing direction of KL divergence between uniform distributions and the output distributions.

DisturbLabel introduced by Xie et al. (2016) imposes the regularization within the loss layer, where it randomly replaces some of the ground truth labels as incorrect values at each training iteration. Its effect is quite similar to LSR that can help to prevent the neural network training from overfitting. The authors have verified the effectiveness of DisturbLabel via several experiments on training image classification tasks.

Recently, many works (Zhang et al., 2018; Bagherinezhad et al., 2018; Goibert & Dohmatob, 2019; Shen et al., 2019; Li et al., 2020b) explored the idea of LSR technique. Ding et al. (2019) extended an adaptive label regularization method, which enables the neural network to use both correctness and incorrectness during training. Pang et al. (2018) used the reverse cross-entropy loss to smooth the classifier's gradients. Wang et al. (2020) proposed a graduated label smoothing method that uses the higher smoothing penalty for high-confidence predictions than that for low-confidence predictions. They found that the proposed method can improve both inference calibration and translation performance for neural machine translation models. By contrast, we will try to understand the power of LSR from an optimization perspective and try to study how and when to use LSR.

## 3 Preliminaries and Notations

We first present some notations. Let $\nabla_{\mathbf{w}} F_{\mathcal{S}}(\mathbf{w})$ denote the gradient of a function $F_{\mathcal{S}}(\mathbf{w})$. When the variable to be taken a gradient is obvious, we use $\nabla F_{\mathcal{S}}(\mathbf{w})$ for simplicity. We use $\| \cdot \|$ to denote the Euclidean norm. Let $\langle \cdot, \cdot \rangle$ be the inner product.

In classification problem, we aim to seek a classifier to map an example $\mathbf{x} \in \mathcal{X}$ onto one of $K$ labels $\mathbf{y} \in \mathcal{Y} \subset \mathbb{R}^K$, where $\mathbf{y} = (y_1, y_2, \ldots, y_K)$ is a one-hot label, meaning that $y_i$ is "1" for the correct class and "0" for the rest. Suppose the example-label pairs are drawn from a distribution $\mathbb{P}$, i.e., $(\mathbf{x}, \mathbf{y}) \sim \mathbb{P} = (\mathbb{P}_{\mathbf{x}}, \mathbb{P}_{\mathbf{y}})$. Let $\mathcal{S} = \{(\mathbf{x}_1, \mathbf{y}_1), \ldots, (\mathbf{x}_n, \mathbf{y}_n)\}$ denotes a set of $n$ examples drawn from $\mathbb{P}$. We denote by $\mathrm{E}_{(\mathbf{x},\mathbf{y})}[\cdot]$ the expectation that takes over a random variable $(\mathbf{x}, \mathbf{y})$. When the randomness is obvious, we write $\mathrm{E}[\cdot]$ for simplicity. Our goal is to learn a prediction function $f(\mathbf{w}; \mathbf{x}) : \mathcal{W} \times \mathcal{X} \to \mathbb{R}^K$ that is as close as possible

to $\mathbf{y}$, where $\mathbf{w} \in \mathcal{W}$ is the parameter and $\mathcal{W}$ is a closed convex set. To this end, we want to minimize the following expected loss under $\mathbb{P}$:

$$\min_{\mathbf{w} \in \mathcal{W}} F_{\mathcal{S}}(\mathbf{w}) := \frac{1}{n} \sum_{i=1}^{n} \ell(\mathbf{y}_i, f(\mathbf{w}; \mathbf{x}_i)) \tag{1}$$

where $\ell : \mathcal{Y} \times \mathbb{R}^K \to \mathbb{R}_+$ is a cross-entropy loss function given by

$$\ell(\mathbf{y}, f(\mathbf{w}; \mathbf{x})) = \sum_{i=1}^{K} -y_i \log \left( \frac{\exp(f_i(\mathbf{w}; \mathbf{x}))}{\sum_{j=1}^{K} \exp(f_j(\mathbf{w}; \mathbf{x}))} \right). \tag{2}$$

The objective function $F_{\mathcal{S}}(\mathbf{w})$ is not convex since $f(\mathbf{w}; \mathbf{x})$ is non-convex in terms of $\mathbf{w}$. To solve the problem (1), one can simply use some iterative methods such as stochastic gradient descent (SGD). Specifically, at each training iteration $t$, SGD updates solutions iteratively by $\mathbf{w}_{t+1} = \mathbf{w}_t - \eta \nabla_{\mathbf{w}} \ell(\mathbf{y}_t, f(\mathbf{w}_t; \mathbf{x}_t))$, where $\eta > 0$ is a learning rate.

Next, we present some notations and assumptions that will be used in the convergence analysis. Throughout this paper, we also make the following assumptions for solving the problem (1).

**Assumption 1.** *Assume the following conditions hold:*

*(i) The stochastic gradient of $F_{\mathcal{S}}(\mathbf{w})$ is unbiased, i.e., $\mathrm{E}_{(\mathbf{x}, \mathbf{y})}[\nabla \ell(\mathbf{y}, f(\mathbf{w}; \mathbf{x}))] = \nabla F_{\mathcal{S}}(\mathbf{w})$, and the variance of stochastic gradient is bounded, i.e., there exists a constant $\sigma^2 > 0$, such that $\mathrm{E}_{(\mathbf{x}, \mathbf{y})} \left[ \|\nabla \ell(\mathbf{y}, f(\mathbf{w}; \mathbf{x})) - \nabla F_{\mathcal{S}}(\mathbf{w})\|^2 \right] = \sigma^2.$*

*(ii) $F_{\mathcal{S}}(\mathbf{w})$ is smooth with an $L$-Lipchitz continuous gradient, i.e., it is differentiable and there exists a constant $L > 0$ such that $\|\nabla F_{\mathcal{S}}(\mathbf{w}) - \nabla F_{\mathcal{S}}(\mathbf{u})\| \leq L\|\mathbf{w} - \mathbf{u}\|, \forall \mathbf{w}, \mathbf{u} \in \mathcal{W}.$*

**Remark.** Assumption 1 (i) and (ii) are commonly used assumptions in the literature of non-convex optimization (Ghadimi & Lan, 2013; Yan et al., 2018; Yuan et al., 2019b; Wang et al., 2019; Li et al., 2020a). Assumption 1 (ii) says the objective function is $L$-smooth, and it has an equivalent expression (Nesterov, 2004) which is $F_{\mathcal{S}}(\mathbf{w}) - F_{\mathcal{S}}(\mathbf{u}) \leq \langle \nabla F_{\mathcal{S}}(\mathbf{u}), \mathbf{w} - \mathbf{u} \rangle + \frac{L}{2}\|\mathbf{w} - \mathbf{u}\|^2, \forall \mathbf{w}, \mathbf{u} \in \mathcal{W}$. For a classification problem, the smoothed label $\mathbf{y}^{\mathrm{LS}}$ is given by

$$\mathbf{y}^{\mathrm{LS}} = (1 - \theta)\mathbf{y} + \theta \widehat{\mathbf{y}}, \tag{3}$$

where $\theta \in (0, 1)$ is the smoothing strength, $\mathbf{y}$ is the one-hot label, $\widehat{\mathbf{y}}$ is an introduced label. For example, one can simply use $\widehat{\mathbf{y}} = \frac{1}{K}$ (Szegedy et al., 2016) for $K$-class problems. Similar to label $\mathbf{y}$, we suppose the label $\widehat{\mathbf{y}}$ is drawn from a distribution $\mathbb{P}_{\widehat{\mathbf{y}}}$. We introduce the variance of stochastic gradient using label $\widehat{\mathbf{y}}$ as follows.

$$\mathrm{E}_{(\mathbf{x}, \widehat{\mathbf{y}})} \left[ \|\nabla \ell(\widehat{\mathbf{y}}, f(\mathbf{w}; \mathbf{x})) - \nabla F_{\mathcal{S}}(\mathbf{w})\|^2 \right] = \widehat{\sigma}^2 := \delta \sigma^2. \tag{4}$$

where $\delta > 0$ is a constant and $\sigma^2$ is defined in Assumption 1 (i). We make several remarks for (4).

**Remark.** (a) We do not require that the stochastic gradient $\nabla \ell(\widehat{\mathbf{y}}, f(\mathbf{w}; \mathbf{x}))$ is unbiased, i.e., it could be $\mathrm{E}[\nabla \ell(\widehat{\mathbf{y}}, f(\mathbf{w}; \mathbf{x}))] \neq \nabla F_{\mathcal{S}}(\mathbf{w})$. (b) The variance $\widehat{\sigma}^2$ is defined based on the label $\widehat{\mathbf{y}}$ rather than the smoothed label $\mathbf{y}^{\mathrm{LS}}$. (c) We do not assume the variance $\widehat{\sigma}^2$ is bounded since $\delta$ could be an arbitrary value, however, we will discuss the different cases of $\delta$ in our analysis. If $\delta \geq 1$, then $\widehat{\sigma}^2 \geq \sigma^2$; while if $0 < \delta < 1$, then $\widehat{\sigma}^2 < \sigma^2$. It is worth mentioning that $\delta$ could be small when an appropriate label is used in the label smoothing. For example, one can smooth labels by using a teacher model (Hinton et al., 2014) or the model's own distribution (Reed et al., 2014). In the first paper of label smoothing (Szegedy et al., 2016) and the following related studies (Müller et al., 2019; Yuan et al., 2019a), researchers consider a uniform distribution over all $K$ classes of labels as the label $\widehat{\mathbf{y}}$, i.e., set $\widehat{\mathbf{y}} = \frac{1}{K}$. Due to the space limitation, **we include more discussions and the evaluations of $\delta$ in real-world applications in Appendix A.1.** For example, we show that $\delta < 1$ for CIFAR-100 in practice.

We now introduce an important property regarding $F_{\mathcal{S}}(\mathbf{w})$, i.e. Polyak-Łojasiewicz (PL) condition (Polyak, 1963). More specifically, the following assumption holds.

---

**Algorithm 1** SGD with Label Smoothing Regularization

---
1: **Initialize**: $\mathbf{w}_0 \in \mathcal{W}$, $\theta \in (0, 1)$, set $\eta$ as the value in Theorem 3.
2: **for** $t = 0, 1, \ldots, T - 1$ **do**
3:     sample $(\mathbf{x}_t, \mathbf{y}_t)$, set $\mathbf{y}_t^{\mathrm{LS}} = (1 - \theta)\mathbf{y}_t + \theta\widehat{\mathbf{y}}_t$
4:     update $\mathbf{w}_{t+1} = \mathbf{w}_t - \eta\nabla_{\mathbf{w}}\ell(\mathbf{y}_t^{\mathrm{LS}}, f(\mathbf{w}_t; \mathbf{x}_t))$
5: **end for**
6: **Output:**   $w_R$, where $R$ is uniformly sampled from $\{0, 1, \ldots, T - 1\}$.

---

**Assumption 2.** *There exists a constant $\mu > 0$ such that $2\mu(F_{\mathcal{S}}(\mathbf{w}) - F_{\mathcal{S}}(\mathbf{w}_{\mathcal{S}}^*)) \leq \|\nabla F_{\mathcal{S}}(\mathbf{w})\|^2, \forall\mathbf{w} \in \mathcal{W}$, where $\mathbf{w}_{\mathcal{S}}^* \in \min_{\mathbf{w}\in\mathcal{W}} F_{\mathcal{S}}(\mathbf{w})$ is a optimal solution.*

**Remark.** This property has been theoretically and empirically observed in training deep neural networks (Allen-Zhu et al., 2019; Yuan et al., 2019b). This condition is widely used to establish convergence in the literature of non-convex optimization, please see (Yuan et al., 2019b; Wang et al., 2019; Karimi et al., 2016; Li & Li, 2018; Charles & Papailiopoulos, 2018; Li et al., 2020a) and references therein.

To measure the convergence of non-convex and smooth optimization problems as in (Nesterov, 1998; Ghadimi & Lan, 2013; Yan et al., 2018), we need the following definition of the first-order stationary point.

**Definition 1** (First-order stationary point). *For the problem of $\min_{\mathbf{w}\in\mathcal{W}} F_{\mathcal{S}}(\mathbf{w})$, a point $\mathbf{w} \in \mathcal{W}$ is called a first-order stationary point if $\|\nabla F_{\mathcal{S}}(\mathbf{w})\| = 0$. Moreover, if $\|\nabla F_{\mathcal{S}}(\mathbf{w})\| \leq \epsilon$, then the point $\mathbf{w}$ is said to be an $\epsilon$-stationary point, where $\epsilon \in (0, 1)$ is a small positive value.*

## 4 TSLA: A Two-Stage Label Smoothing Algorithm

In this section, we present our main method with its convergence analysis. As a warm-up, we first show the convergence analysis of SGD with LSR to understand LS from theoretical side. Then we will introduce the proposed TSLA and study its convergence results both in training error (optimization) and testing error (generalization).

### 4.1 Convergence Analysis of SGD with LSR

To understand LSR from the optimization perspective, we consider SGD with LSR in Algorithm 1 for the sake of simplicity. The only difference between Algorithm 1 and standard SGD is the use of the output label for constructing a stochastic gradient. The following theorem shows that Algorithm 1 converges to an approximate stationary point in expectation under some conditions. We include its proof in Appendix A.3.

**Theorem 3.** *Under Assumption 1, run Algorithm 1 with $\eta = \frac{1}{L}$ and $\theta = \frac{1}{1+\delta}$, then $\mathrm{E}_R[\|\nabla F_{\mathcal{S}}(\mathbf{w}_R)\|^2] \leq \frac{2F_{\mathcal{S}}(\mathbf{w}_0)}{\eta T} + 2\delta\sigma^2$, where $R$ is uniformly sampled from $\{0, 1, \ldots, T - 1\}$. Furthermore, given a target accuracy level $\epsilon$, we have the following two results.*
*(1) when $\delta \leq \frac{\epsilon^2}{4\sigma^2}$, if we set $T = \frac{4F_{\mathcal{S}}(\mathbf{w}_0)}{\eta\epsilon^2}$, then Algorithm 1 converges to an $\epsilon$-stationary point in expectation, i.e., $\mathrm{E}_R[\|\nabla F_{\mathcal{S}}(\mathbf{w}_R)\|^2] \leq \epsilon^2$. The total sample complexity is $T = O\left(\frac{1}{\epsilon^2}\right)$.*
*(2) when $\delta > \frac{\epsilon^2}{4\sigma^2}$, if we set $T = \frac{F_{\mathcal{S}}(\mathbf{w}_0)}{\eta\delta\sigma^2}$, then Algorithm 1 does not converge to an $\epsilon$-stationary point, but we have $\mathrm{E}_R[\|\nabla F_{\mathcal{S}}(\mathbf{w}_R)\|^2] \leq 4\delta\sigma^2 \leq O(\delta)$.*

**Remark.** We observe that the variance term is $2\delta\sigma^2$, instead of $\eta L\sigma^2$ for standard analysis of SGD without LSR (i.e., $\theta = 0$, please see the detailed analysis of Theorem 1 in Appendix A.4). For the convergence analysis, the difference between SGD with LSR and SGD without LSR is that $\nabla\ell(\widehat{\mathbf{y}}, f(\mathbf{w}; \mathbf{x}))$ is not an unbiased estimator of $\nabla F_{\mathcal{S}}(\mathbf{w})$ when using LSR. The convergence behavior of Algorithm 1 heavily depends on the parameter $\delta$. When $\delta$ is small enough, say $\delta \leq O(\epsilon^2)$ with a small positive value $\epsilon \in (0, 1)$, then Algorithm 1 converges to an $\epsilon$-stationary point with the total sample complexity of $O\left(\frac{1}{\epsilon^2}\right)$. Recall that the total sample complexity of standard SGD without LSR for finding an $\epsilon$-stationary point is $O\left(\frac{1}{\epsilon^4}\right)$ ((Ghadimi & Lan, 2016; Ghadimi et al., 2016), please also see the detailed analysis of Theorem 1 in Appendix A.4). The convergence result shows that if we could find a label $\widehat{\mathbf{y}}$ that has a reasonably small amount of $\delta$, we will

**Algorithm 2** The TSLA algorithm

---

1: **Initialize**: $\mathbf{w}_0 \in \mathcal{W}$, $T_1$, $\theta \in (0,1)$, $\eta_1, \eta_2 > 0$
    // First stage: SGD with LSR
2: **for** $t = 0, 1, \ldots, T_1 - 1$ **do**
3:     set $\mathbf{y}_t^{\text{LS}} = (1-\theta)\mathbf{y}_t + \theta\widehat{\mathbf{y}}_t$
4:     update $\mathbf{w}_{t+1} = \mathbf{w}_t - \eta_1 \nabla \ell(\mathbf{y}_t^{\text{LS}}, f(\mathbf{w}_t; \mathbf{x}_t))$
5: **end for**
    // Second stage: SGD without LSR
6: **for** $t = T_1, 1, \ldots, T_1 + T_2 - 1$ **do**
7:     update $\mathbf{w}_{t+1} = \mathbf{w}_t - \eta_2 \nabla \ell(\mathbf{y}_t, f(\mathbf{w}_t; \mathbf{x}_t))$.
8: **end for**
9: **Output:**     $\mathbf{w}_R$, where $R$ is uniformly sampled from $\{T_1, \ldots, T_1 + T_2 - 1\}$.

---

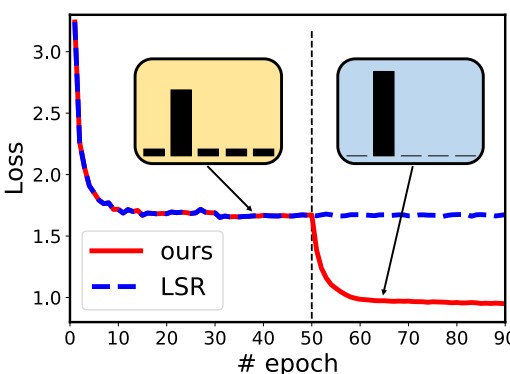

Figure 1: Loss on ResNet-18 over CUB-2011.

be able to reduce sample complexity for training a machine learning model from $O\left(\frac{1}{\epsilon^4}\right)$ to $O\left(\frac{1}{\epsilon^2}\right)$. Thus, the reduction in variance will happen when an appropriate label smoothing with $\delta \in (0,1)$ is introduced. We will find in the empirical evaluations that different label $\widehat{\mathbf{y}}$ lead to different performances and an appropriate selection of label $\widehat{\mathbf{y}}$ has a better performance (see the performances of LSR and LSR-pre in Table 2). On the other hand, when the parameter $\delta$ is large such that $\delta > \Omega(\epsilon^2)$, that is to say, if an inappropriate label smoothing is used, then Algorithm 1 does not converge to an $\epsilon$-stationary point, but it converges to a worse level of $O(\delta)$.

## 4.2 The TSLA Algorithm

Previous subsection have shown that LSR could not find an $\epsilon-$stationary point under some situations. These motivate us to investigate a strategy that combines the algorithm with and without LSR during the training progress. Let think in this way, one possible scenario is that training one-hot label is "easier" than training smoothed label. Taking the cross entropy loss in (2) for an example, one need to optimize a single loss function $-\log\left(\exp(f_k(\mathbf{w};\mathbf{x}))/\sum_{j=1}^K \exp(f_j(\mathbf{w};\mathbf{x}))\right)$ when one-hot label (e.g, $y_k = 1$ and $y_i = 0$ for all $i \neq k$) is used, but need to optimize all $K$ loss functions $-\sum_{i=1}^K \mathbf{y}_i^{\text{LS}} \log\left(\exp(f_i(\mathbf{w};\mathbf{x}))/\sum_{j=1}^K \exp(f_j(\mathbf{w};\mathbf{x}))\right)$ when smoothed label (e.g., $\mathbf{y}^{\text{LS}} = (1-\theta)\mathbf{y} + \theta\frac{1}{K}$ so that $y_k^{\text{LS}} = 1 - (K-1)\theta/K$ and $y_i^{\text{LS}} = \theta/K$ for all $i \neq k$) is used. Nevertheless, training deep neural networks is gradually focusing on hard examples with the increase of training epochs. It seems that training with smoothed label in the late epochs makes the learning progress more difficult. In addition, after LSR, we focus on optimizing the overall distribution that contains the minor classes, which are probably not important at the end of training progress. Please see the blue dashed line (trained by SGD with LSR) in Figure 1[1], and it shows that there is almost no change in loss after 30 epochs. One question is whether LSR helps at the early training epochs but it has less (even negative) effect during the later training epochs? This question encourages us to propose and analyze a simple strategy with LSR dropping that switches a stochastic algorithm with LSR to the algorithm without LSR.

A natural choice could be just dropping off LSR after certain epochs. For example, one can run SGD with LSR in the first 50 epochs and then run SGD without LSR after that (see red solid line in Figure 1). From Figure 1, we can see that the drop of LSR after 50 epochs can further decrease the loss function.

With this in mind in this subsection, we propose a generic framework that consists of two stages, wherein the first stage it runs SGD with LSR for $T_1$ iterations and the second stage it runs SGD without LSR up to $T_2$ iterations. This framework is referred to as **T**wo-**S**tage **LA**bel smoothing (TSLA) algorithm, whose updating details are presented in Algorithm 2. Although SGD is considered as the pipeline (Line 4 and Line 8 in Algorithm 2) in the convergence analysis, in practice, SGD can be replaced by any stochastic algorithms such as momentum SGD (Polyak, 1964), Stochastic Nesterov's Accelerated Gradient (Nesterov,

---

[1]The blue dashed line is trained by SGD with LSR. The red solid line is trained by SGD with LSR for the first 50 epochs and trained by SGD without LSR (i.e., using one-hot label) for the last 40 epochs.

1983), and adaptive algorithms including ADAGRAD (Duchi et al., 2011), RMSProp (Hinton et al., 2012a), AdaDelta (Zeiler, 2012), Adam (Kingma & Ba, 2015), Nadam (Dozat, 2016) and AMSGRAD (Reddi et al., 2018). In this paper, we will not study the theoretical guarantees and empirical evaluations of other optimizers, which can be considered as future work. Please note that the algorithm can use different learning rates $\eta_1$ and $\eta_2$ during the two stages. The last solution of the first stage will be used as the initial solution of the second stage. If $T_1 = 0$, then TSLA reduces to the baseline, i.e., SGD without LSR; while if $T_2 = 0$, TSLA becomes to LSR method, i.e., SGD with LSR.

### 4.3 Optimization Result of TSLA

In this subsection, we will give the convergence result of the proposed TSLA algorithm. For simplicity, we use SGD as the subroutine algorithm $\mathcal{A}$ in the analysis. The convergence result in the following theorem shows the power of LSR from the optimization perspective. Its proof is presented in Appendix A.5. It is easy to see from the proof that by using the last output of the first stage as the initial point of the second stage, TSLA can enjoy the advantage of LSR in the second stage with an improved convergence.

**Theorem 4.** *Under Assumptions 1, 2, suppose $\sigma^2 \delta / \mu \leq F(\mathbf{w}_0)$, run Algorithm 2, $\theta = \frac{1}{1+\delta}$, $\eta_1 = \frac{1}{L}$, $T_1 = \log\left(\frac{2\mu F(\mathbf{w}_0)(1+\delta)}{2\delta\sigma^2}\right)/(\eta_1 \mu)$, $\eta_2 = \frac{\epsilon^2}{2L\sigma^2}$ and $T_2 = \frac{8\delta\sigma^2}{\mu \eta_2 \epsilon^2}$, then $\mathrm{E}_R[\|\nabla F_{\mathcal{S}}(\mathbf{w}_R)\|^2] \leq \epsilon^2$, where $R$ is uniformly sampled from $\{T_1, \ldots, T_1 + T_2 - 1\}$.*

**Remark.** It is obvious that the learning rate $\eta_2$ in the second stage is roughly smaller than the learning rate $\eta_1$ in the first stage, which matches the widely used stage-wise learning rate decay scheme in training neural networks. To explore the total sample complexity of TSLA, we consider different conditions on $\delta$. For a fixed the target convergence level $\epsilon \in (0, 1)$, let us discuss the total sample complexities of finding $\epsilon$-stationary points for SGD with TSLA (TSLA), SGD with LSR (LSR), and SGD without LSR (baseline), where we only consider the orders of the complexities but ignore all constants. When $\Omega(\epsilon^2) < \delta < 1$, LSR does not converge to an $\epsilon$-stationary point, while TSLA reduces sample complexity from $O\left(\frac{1}{\epsilon^4}\right)$ to $O\left(\frac{\delta}{\epsilon^4}\right)$, compared to the baseline. When $\delta < O(\epsilon^2)$, the total complexity of TSLA is between $\log(1/\epsilon)$ and $1/\epsilon^2$, which is always better than LSR and the baseline. In summary, TSLA achieves the best total sample complexity by enjoying the good property of an appropriate label smoothing (i.e., when $0 < \delta < 1$). However, when $\delta \geq 1$, baseline has better convergence than TSLA, meaning that the selection of label $\widehat{\mathbf{y}}$ is not appropriate. Since $T_1$ contains unknown parameters, it is difficulty to know its ground-truth value. However, we can tune different values in practice.

### 4.4 Generalization Result of TSLA

In this subsection, we study the generalization result of TSLA. First, we give some notations, where most of them are followed by (Hardt et al., 2015; Yuan et al., 2019b). Let $\mathbf{w}_{\mathcal{S}} = \mathcal{A}(\mathcal{S})$ be a solution that generated by a random algorithm $\mathcal{A}$ based on dataset $\mathcal{S}$. Recall that problem (1) is called empirical risk minimization in the literature, and the true risk minimization is given by

$$\min_{\mathbf{w} \in \mathcal{W}} F(\mathbf{w}) := \mathrm{E}_{(\mathbf{x}, \mathbf{y})} \left[ \ell(\mathbf{y}, f(\mathbf{w}; \mathbf{x})) \right]. \tag{5}$$

Then the testing error is dedined as

$$\mathrm{E}_{\mathcal{A}, \mathcal{S}}[F(\mathbf{w}_{\mathcal{S}})] - \mathrm{E}_{\mathcal{S}}[F_{\mathcal{S}}(\mathbf{w}_{\mathcal{S}}^*)] \tag{6}$$

We notice that there are several works (Hardt et al., 2015; Yuan et al., 2019b) study the testing error result of SGD for non-convex setting under different conditions such as bounded stochastic gradient $\|\nabla_{\mathbf{w}} \ell(\mathbf{y}, f(\mathbf{w}; \mathbf{x}))\| \leq G$ and decaying learning rate $\eta_t \leq \frac{c}{t}$ with a constant $c > 0$, where $t$ is the optimization iteration. In this paper, we are not interested in establishing fast rate under different conditions, but we want to explore the generalization ability of TSLA with the fewest possible modifications when building a bridge between theory and practice. For example, weight decay is a widely used trick when training deep neural networks. With the use of weight decay, the empirical risk minimization in practice becomes $\min_{\mathbf{w} \in \mathcal{W}} \{\widehat{F}_{\mathcal{S}}(\mathbf{w}) := F_{\mathcal{S}}(\mathbf{w}) + \frac{\lambda}{2} \|\mathbf{w}\|^2\}$. We present the ERB of TSLA in the following Theorem 5, whose proofs can be found in the Appendix A.6.

**Theorem 5.** *Under Assumption 1, assume that $\ell(\mathbf{y}, f(\mathbf{w}, \mathbf{x}))$ is $L$-smooth and $B$-Lipschitz, suppose $\widehat{F}_{\mathcal{S}}(\mathbf{w})$ satisfies Assumption 2 and $\min_{\mathbf{w} \in \mathcal{W}} \widehat{F}_{\mathcal{S}}(\mathbf{w}) \leq F_{\mathcal{S}}(\mathbf{w}_{\mathcal{S}}^*) + \frac{\lambda}{2}\|\mathbf{w}_t\|^2$ with $\lambda = 2L$, where $\mathbf{w}_t$ is the intermediate solution in the second stage of Algorithm 2 by running with $\theta = \frac{1}{1+\delta}$, $\eta_1 < \frac{1}{3L}$, $T_1 = \log\left(\frac{2\mu F_{\mathcal{S}}(\mathbf{w}_0)(1+\delta)}{2\delta\sigma^2}\right)/(\eta_1 \mu)$, $\eta_2 = O(1/\sqrt{n})$ and $T_2 = O(\delta n)$ then the testing error $\mathrm{E}_{R,\mathcal{A},\mathcal{S}}[F(\mathbf{w}_R)] - \mathrm{E}_{\mathcal{S}}[F_{\mathcal{S}}(\mathbf{w}_{\mathcal{S}}^*)] \leq O(1/\sqrt{n})$, where $\mathcal{A}$ is TSLA.*

**Remark.** Theorem 5 shows that in order to have an bound of testing error in the order of $O(1/\sqrt{n})$, TSLA needs to run $T = O(\delta n)$ iterations. For the standard analysis of SGD with/without LSR (please see the remarks in Appendix A.6) under the same conditions, SGD without LSR needs to run $T = O(n)$ iterations to obtain an $O(1/\sqrt{n})$ bound of testing error. Thus, if $\delta$ is small enough, the testing error of TSLA is better than that of SGD without LSR, otherwise, SGD without LSR has better testing error than TSLA. For SGD with LSR, when $\delta > \Omega(1/\sqrt{n})$, the bound of testing error for LSR can not be bounded by $O(1/\sqrt{n})$; when $\delta \leq \Omega(1/\sqrt{n})$, the bound for LSR can be bounded by $O(1/\sqrt{n})$ in $T = O(\sqrt{n})$ iterations. This shows that TSLA bas better testing error than LSR.

## 5 Experiments

To further evaluate the performance of the proposed TSLA method, we trained deep neural networks on three benchmark data sets, CIFAR-100 (Krizhevsky & Hinton, 2009), Stanford Dogs (Khosla et al., 2011) and CUB-2011 (Wah et al., 2011), for image classification tasks. CIFAR-100 [2] has 50,000 training images and 10,000 testing images of 32×32 resolution with 100 classes. Stanford Dogs data set [3] contains 20,580 images of 120 breeds of dogs, where 100 images from each breed is used for training. CUB-2011 [4] is a birds image data set with 11,788 images of 200 birds species. The ResNet-18 model (He et al., 2016) is applied as the backbone in the experiments.

We compare the proposed TSLA incorporing with SGD (TSLA) with two baselines, SGD with LSR (LSR) and SGD without LSR (baseline). The mini-batch size of training instances for all methods is 256 as suggested by (He et al., 2019; 2016). The momentum parameter is fixed as 0.9. We will include more details of experimental settings in Appendix A.1.

### 5.1 Stanford Dogs and CUB-2011

We separately train ResNet-18 (He et al., 2016) up to 90 epochs over two data sets Stanford Dogs and CUB-2011. We use weight decay with the parameter value of $10^{-4}$. For all algorithms, the initial learning rates for FC are set to be 0.1, while that for the pre-trained backbones are 0.001 and 0.01 for Standford Dogs and CUB-2011, respectively. The learning rates are divided by 10 every 30 epochs. For LSR, we fix the value of smoothing strength $\theta = 0.4$ for the best performance, and the label $\widehat{\mathbf{y}}$ used for label smoothing is set to be a uniform distribution over all $K$ classes, i.e., $\widehat{\mathbf{y}} = \frac{1}{K}$. The same values of the smoothing strength $\theta$ and the same $\widehat{\mathbf{y}}$ are used during the first stage of TSLA. For TSLA, we drop off the LSR (i.e., let $\theta = 0$) after $s$ epochs during the training process, where $s \in \{20, 30, 40, 50, 60, 70, 80\}$.

We first report the highest top-1 and top-5 accuracy on the testing data sets for different methods. All top-1 and top-5 accuracy are averaged over 5 independent random trails with their standard deviations. The results of the comparison are summarized in Table 1, where the notation "TSLA($s$)" means that the TSLA algorithm drops off LSR after epoch $s$. It can be seen from Table 1 that under an appropriate hyperparameter setting the models trained using TSLA outperform that trained using LSR and baseline, which supports the convergence result in Section 4.2. We notice that the best top-1 accuracy of TSLA are TSLA(40) and TSLA(50) for Stanford Dogs and CUB-2011, respectively, meaning that the performance of TSLA($s$) is not monotonic over the dropping epoch $s$. For CUB-2011, the top-1 accuracy of TSLA(20) is smaller than that of LSR. This result matches the convergence analysis of TSLA showing that it can not drop off LSR too early. For top-5 accuracy, we found that TSLA(80) is slightly worse than baseline. This

---

[2] https://www.cs.toronto.edu/~kriz/cifar.html
[3] http://vision.stanford.edu/aditya86/ImageNetDogs/
[4] http://www.vision.caltech.edu/visipedia/CUB-200.html

Table 1: Comparison of Testing Accuracy for Different Methods (mean ± standard deviation, in %).

| Algorithm* | Stanford Dogs | | CUB-2011 | |
| | Top-1 accuracy | Top-5 accuracy | Top-1 accuracy | Top-5 accuracy |
|---|---|---|---|---|
| baseline | $82.31 \pm 0.18$ | $97.76 \pm 0.06$ | $75.31 \pm 0.25$ | $93.14 \pm 0.31$ |
| LSR | $82.80 \pm 0.07$ | $97.41 \pm 0.09$ | $76.97 \pm 0.19$ | $92.73 \pm 0.12$ |
| TSLA(20) | $83.15 \pm 0.02$ | $97.91 \pm 0.08$ | $76.62 \pm 0.15$ | $93.60 \pm 0.18$ |
| TSLA(30) | $83.89 \pm 0.16$ | $98.05 \pm 0.08$ | $77.44 \pm 0.19$ | $93.92 \pm 0.16$ |
| TSLA(40) | $\mathbf{83.93} \pm 0.13$ | $98.03 \pm 0.05$ | $77.50 \pm 0.20$ | $93.99 \pm 0.11$ |
| TSLA(50) | $83.91 \pm 0.15$ | $\mathbf{98.07} \pm 0.06$ | $\mathbf{77.57} \pm 0.21$ | $93.86 \pm 0.14$ |
| TSLA(60) | $83.51 \pm 0.11$ | $97.99 \pm 0.06$ | $77.25 \pm 0.29$ | $\mathbf{94.43} \pm 0.18$ |
| TSLA(70) | $83.38 \pm 0.09$ | $97.90 \pm 0.09$ | $77.21 \pm 0.15$ | $93.31 \pm 0.12$ |
| TSLA(80) | $83.14 \pm 0.09$ | $97.73 \pm 0.07$ | $77.05 \pm 0.14$ | $93.05 \pm 0.08$ |

*TSLA($s$): TSLA drops off LSR after epoch $s$.

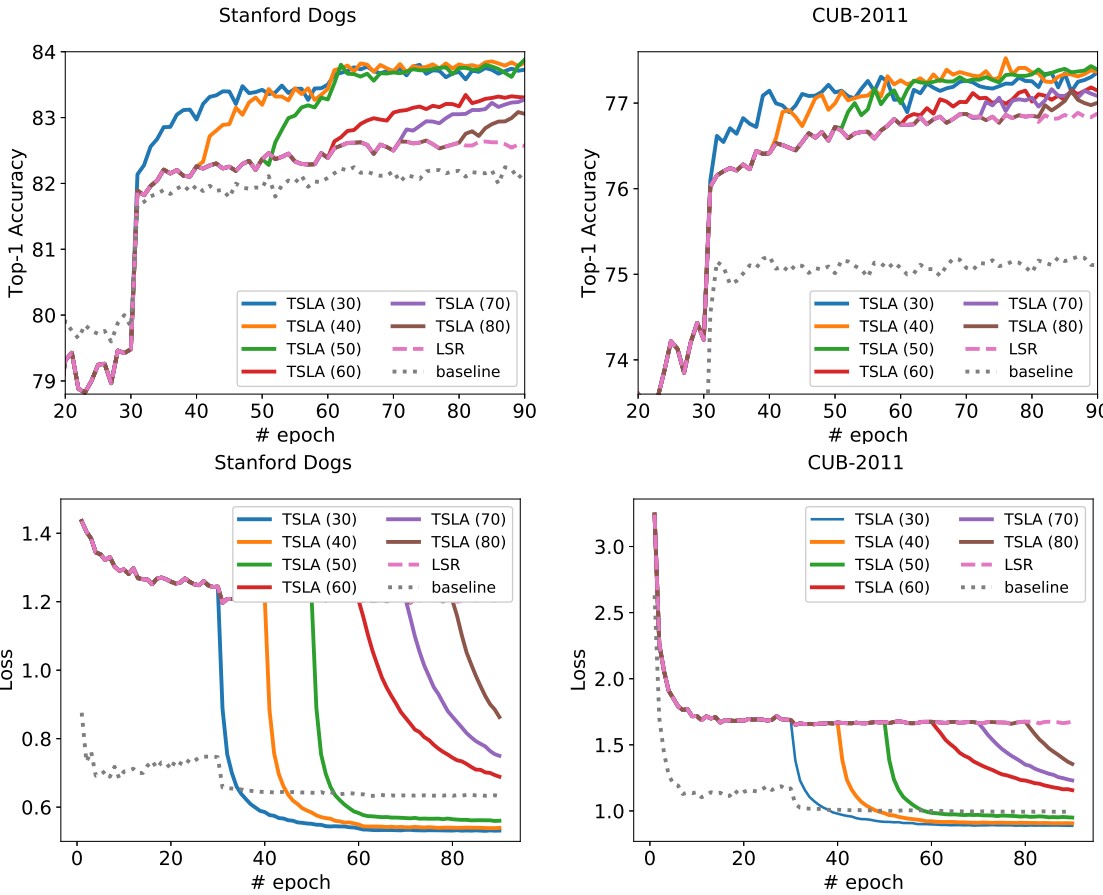

Figure 2: Testing Top-1 and Loss on ResNet-18 over Stanford Dogs and CUB-2011. TSLA($s$) means TSLA drops off LSR after epoch $s$.

is because of dropping LSR too late so that the update iterations (i.e., $T_2$) in the second stage of TSLA is too small to converge to a good solution. We also observe that LSR is better than baseline regarding top-1 accuracy but the result is opposite as to top-5 accuracy.

We then plot the averaged top-1 accuracyand averaged loss among 5 trails of different methods in Figure 2. We remove the results for TSLA(20) since it dropped off LSR too early as mentioned before. The figure shows TSLA improves the top-1 testing accuracy immediately once it drops off LSR. Although TSLA may not converge if it drops off LSR too late, see TSLA(60), TSLA(70), and TSLA(80) from the third column of Figure 2, it still has the best performance compared to LSR and baseline. TSLA(30), TSLA(40), and TSLA(50) can converge to lower objective levels, comparing to LSR and baseline.

## 5.2 CIFAR-100

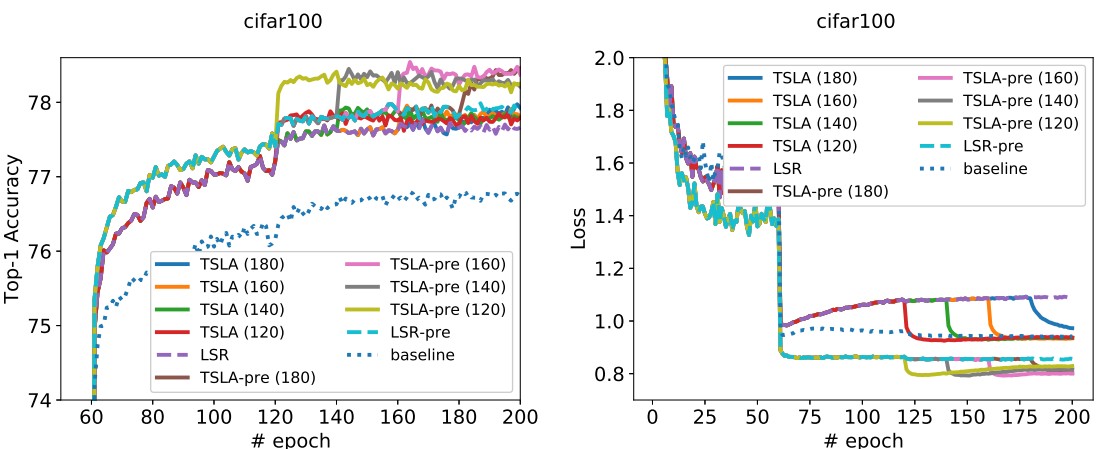

Figure 3: Testing Top-1 and Loss on ResNet-18 over CIFAR-100. TSLA($s$)/TSLA-pre($s$) meansTSLA/TSLA-pre drops off LSR/LSR-pre after epoch $s$.

The total epochs of training ResNet-18 (He et al., 2016) on CIFAR-100 is set to be 200.The weight decay with the parameter value of $5 \times 10^{-4}$ is used. We use 0.1 as the initial learning rates for all algorithms and divide them by 10 every 60 epochs as suggested in (He et al., 2016; Zagoruyko & Komodakis, 2016). For LSR and the first stage of TSLA, the value of smoothing strength $\theta$ is fixed as $\theta = 0.1$, which shows the best performance for LSR. We use two different labels $\hat{\mathbf{y}}$ to smooth the one-hot label, the uniform distribution over all labels and the distribution predicted by an ImageNet pre-trained model which is downloaded directly from PyTorch [5] (Paszke et al., 2019). For TSLA, we try to drop off the LSR after $s$ epochs during the training process, where $s \in \{120, 140, 160, 180\}$.

All top-1 and top-5 accuracy on the testing data set are averaged over 5 independent random trails with their standard deviations. We summarize the results in Table 2, where LSR-pre and TSLA-pre indicate LSR and TSLA use the label $\hat{\mathbf{y}}$ based on the ImageNet pre-trained model. The results show that LSR-pre/TSLA-pre has a better performance than LSR/TSLA. The reason might be that the pre-trained model-based prediction is closer to the ground truth than the uniform prediction and it has lower variance (smaller $\delta$). Then, TSLA (LSR) with such pre-trained model-based prediction converges faster than TSLA (LSR) with uniform prediction, which verifies our theoretical findings in Section 4.2 (Section 4.1). This observation also empirically tells us the selection of the prediction function $\hat{\mathbf{y}}$ used for smoothing label is the key to the success of TSLA as well as LSR. Among all methods, the performance of TSLA-pre is the best. For top-1 accuracy, TSLA-pre(160) outperforms all other algorithms, while for top-5 accuracy, TSLA-pre(180) has the best performance.

---

[5]https://pytorch.org/docs/stable/torchvision/models.html

Table 2: Comparison of Testing Accuracy for Different Methods (mean ± standard deviation, in %).

| Algorithm* | CIFAR-100 | |
| | Top-1 accuracy | Top-5 accuracy |
| --- | --- | --- |
| baseline | 76.87 ± 0.04 | 93.47 ± 0.15 |
| LSR | 77.77 ± 0.18 | 93.55 ± 0.11 |
| TSLA(120) | 77.92 ± 0.21 | 94.13 ± 0.23 |
| TSLA(140) | 77.93 ± 0.19 | 94.11 ± 0.22 |
| TSLA(160) | 77.96 ± 0.20 | 94.19 ± 0.21 |
| TSLA(180) | 78.04 ± 0.27 | 94.23 ± 0.15 |
| LSR-pre | 78.07 ± 0.31 | 94.70 ± 0.14 |
| TSLA-pre(120) | 78.34 ± 0.31 | 94.68 ± 0.14 |
| TSLA-pre(140) | 78.39 ± 0.25 | 94.73 ± 0.11 |
| TSLA-pre(160) | **78.55** ± 0.28 | 94.83 ± 0.08 |
| TSLA-pre(180) | 78.53 ± 0.23 | **94.96** ± 0.23 |

*TSLA($s$)/TSLA-pre($s$): TSLA/TSLA-pre drops off LSR/LSR-pre after epoch $s$.

Table 3: Comparison of Testing Top-1 Accuracy for Different $\theta$ of LSR and TSLA (in %).

| $\theta$ | 0.2 | 0.4 | 0.9 |
| --- | --- | --- | --- |
| LSR | 77.75 | 77.72 | 76.40 |
| TSLA(120) | 77.92 | 77.68 | 76.05 |
| TSLA(140) | 78.06 | 77.61 | 76.27 |
| TSLA(160) | 78.09 | 77.54 | 76.37 |
| TSLA(180) | 78.10 | 77.89 | 76.60 |

*TSLA($s$): TSLA drops off LSR after epoch $s$.

Finally, we observe from Figure 3 that both TSLA and TSLA-pre converge, while TSLA-pre converges to the lowest objective value. Similarly, the results of top-1 accuracy show the improvements of TSLA and TSLA-pre at the point of dropping off LSR.

We conduct an **ablation study** for the smoothing parameter $\theta$ and the dropping epoch $s$ in TSLA($s$). We follow the same settings in Subsection 5.2 but use different values of $\theta$ in LSR and TSLA. Specifically, $\theta \in \{0.2, 0.4, 0.9\}$. We use the uniform distribution over all labels to smooth the one-hot label. The results are summarized in Table 3, showing that the different values of $\theta$ and $s$ can affect the performances of LSR and TSLA. With the increase of $\theta$, the performances become worse. However, with different values of $\theta$, TSLA can always outperform LSR when an appropriate dropping epoch $s$ is selected. Besides, TSLA(180) has the best performance for each value of $\theta$.

## 6 Conclusions

In this paper, we have studied the power of LSR in training deep neural networks by analyzing SGD with LSR in different non-convex optimization settings. The convergence results show that an appropriate LSR with reduced label variance can help speed up the convergence. We have proposed a simple and efficient strategy so-called TSLA that can incorporate many stochastic algorithms. The basic idea of TSLA is to switch the training from smoothed label to one-hot label. Integrating TSLA with SGD, we observe from its improved convergence result that TSLA benefits from LSR in the first stage and essentially converges faster in the second stage. Our result also show that TSLA has better testing error than LSR in theory. Throughout extensive experiments, we have shown that TSLA improves the generalization accuracy of deep models on benchmark data sets.

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
