# OpenReview forum: "Towards Understanding Label Smoothing"
_TMLR — Rejected by TMLR_

### Review · Reviewer_W8Js · 2024-02-24

**Summary Of Contributions:**

This paper proposes an algorithm to improve the performance of label smoothing in optimization. Theories and experiments are provided to support their arguments.

**Audience:**

Yes

**Claims And Evidence:**

Yes

**Requested Changes:**

Please see the above comments in weaknesses.

**Strengths And Weaknesses:**

This paper is clear and easy to understand.

One major issue is that some of the writing is not precise. While the author claims that they "theoretically explain why an appropriate LSR can help speed up the convergence", their theory is not complete: Theorem 3 provides the convergence upper bound of LSR, but does not provide lower bound for vanilla method. Only when SLR upper bound is faster than the vanilla lower bound, can we formally claim that LSR is proven to be better. A similar issue appears in the proposed two-stage method, and a lower bound result is needed to do the comparison between the proposed method and vanilla methods, for both algorithm convergence and generalization.

The authors may either provide more theoretical results, or adjust their abstract and introduction to make their contribution description more precise.

Another concern is the experiment: there are comparisons among vanilla method, LSR, and the proposed method. The authors need to conduct ablation study on other factors in the training, e.g., adjust learning rate, momentum, weight decay, dropout, or other methods in literature. Although in some experiments the improvement is around 2.5%, it is possible that the benchmark algorithm is not powerful enough.

If the authors can work on the lower bound result, one further suggestion is to conduct some simulation study based on the scenarios in the lower bound results to show the effectiveness of the proposed method.

---

### Review · Reviewer_9rkV · 2024-03-06

**Summary Of Contributions:**

The paper tries to develop a theory that explains why and when label smoothing can help. The paper analyzes in a K-class classification setting with (label-smoothed) cross-entropy loss, and make assumptions on function smoothness and so on. The paper also provides experimental results to support their theoretical findings.

**Audience:**

No

**Broader Impact Concerns:**

/

**Claims And Evidence:**

No

**Requested Changes:**

Address the two main general points mentioned above.

**Strengths And Weaknesses:**

Overall, the paper seem questionable. Two main points:
1. The writing is quite poor. The wordings and explanations are very inaccurate in many places starting from the abstract. For example, "essentially converges faster" -- what does this mean? "However, these regularization techniques conduct on the hidden activations or weights of a neural network" -- data augmentation does not fall into this description. These are just few out of many examples that I could find.
2. The assumptions/notations do not make sense to me. If we just look at assumption I, in I-(i), the paper assumes the gradient is unbiased, the left hand side is the expectation over data distribution, while the right hand side is average gradient of n samples -- how can we assume these two to be equal? I also don't understand why \sigma^2 does not depend on w, the gradient is obviously a function of the parameter and the expectation is taken over all the samples. All results based on these assumptions are questionable as a result.

---

### Review · Reviewer_3QeH · 2024-03-10

**Summary Of Contributions:**

The paper presents a detailed analysis of the effects of Stochastic Gradient Descent (SGD) with label smoothing on convergence rates and generalization performance within deep learning frameworks. A novel two-stage label smoothing algorithm (TLSA) is introduced, aiming to enhance both convergence speed and generalization abilities. To substantiate the proposed algorithm's efficacy, a series of experiments are conducted across various settings.

**Audience:**

Yes

**Claims And Evidence:**

No

**Requested Changes:**

See above

**Strengths And Weaknesses:**

Strengths

Relevance and Organization: The study addresses a significant and practical aspect of deep learning optimization, focusing on label smoothing's role in SGD. The paper is well-structured, facilitating comprehension of the methodology, experimental setup, and implications of the findings.

Empirical Validation: Through experiments, the paper successfully demonstrates the improved performance of TLSA over traditional SGD methods for the tasks considered. These results are significant, underscoring TLSA's potential in enhancing deep learning models' performance.

Weaknesses

Theoretical Depth: The theoretical analysis primarily extends classical SGD convergence analysis without introducing novel proof techniques. This approach limits the theoretical contribution of the work, as it does not offer new insights into the convergence behavior of SGD with label smoothing.

Dependency on Parameters: The effectiveness of the proposed method is notably dependent on the choice of the parameter
δ, yet the paper does not provide a clear methodology for its computation. This omission could hinder the applicability of TLSA in practice.

Diversity of Models and Datasets: The experimental evaluation predominantly utilizes the ResNet-18 model and relatively simple tasks. Expanding the scope to include more complex models (such as ResNet-50) and challenging datasets (like ImageNet) would provide a more comprehensive understanding of TLSA's performance and applicability across various deep learning scenarios.

Tuning efforts: the two TSLA algorithm additionally introduce the switch epoch which heavily affects the final result and  needs additional tuning efforts.

---

### Decision · Action_Editor_4MBD · 2024-05-27

**Recommendation:** Reject

**Comment:**

The paper proposed a new algorithm to improve the performance of label smoothing in optimization. Theory and experiments are provided to demonstrate the performance of the proposed method. However, the presentation of the paper is poor, which makes it difficult to precisely understand the results of the paper. The authors did not participate during the rebuttal period to clarify the reviewers' concerns and questions, hence losing the opportunity to improve the paper. Thus, I will not recommend to accept the paper.

**Audience:**

Yes. The topic would be interesting for a large set of TMLR audience.

**Claims And Evidence:**

No. The presentation of the paper is not accurate enough to support the claims.